# Oral Charcoal Adsorbents Attenuate Neointima Formation of Arteriovenous Fistulas

**DOI:** 10.3390/toxins12040237

**Published:** 2020-04-08

**Authors:** Yu-Chung Shih, Chih-Cheng Wu, Shen-Chih Wang, Jun-Yang Liou, Po-Hsun Huang, Der-Cherng Tarng

**Affiliations:** 1Division of Plastic and Reconstructive Surgery, Department of Surgery, Taipei Veterans General Hospital, Taipei 11217, Taiwan; rogerstonekimo@gmail.com; 2Institute of Clinical Medicine, National Yang-Ming University, Taipei 11221, Taiwan; dctarng@gmail.com; 3Cardiovascular Center, National Taiwan University Hospital, Hsinchu Branch, Hsinchu 30059, Taiwan; 4College of Medicine, National Taiwan University, Taipei 10051, Taiwan; 5Institute of Biomedical Engineering, National Tsing-Hua University, Hsinchu 30013, Taiwan; 6Institute of Cellular and System Medicine, National Health Research Institutes, Zhunan 35053, Taiwan; jliou@nhri.org.tw; 7Department of Anesthesiology, Taipei Veterans General Hospital, Taipei 11217, Taiwan; akkwang@gmail.com; 8Department of Critical Care Medicine, Taipei Veteran Hospital, Taipei 11217, Taiwan; 9Institute of Physiology, National Yang-Ming University, Taipei 11221, Taiwan; 10Division of Nephrology, Department of Medicine, Taipei Veterans General Hospital, Taipei 11217, Taiwan

**Keywords:** arteriovenous fistula, charcoal adsorbent, indoxyl sulfate, neointima, uremic toxin

## Abstract

Chronic kidney disease (CKD) accelerates the development of neointima formation at the anastomosis site of arteriovenous (AV) fistulas. Accumulation of certain uremic toxins has a deleterious effect on the cardiovascular system. The oral charcoal adsorbent, AST-120, reduces circulating and tissue uremic toxins, but its effect on neointima formation at an AV fistula is unknown. To understand the effect of CKD and AST-120 on neointima formation, we created AV fistulas (common carotid artery to the external jugular vein in an end-to-side anastomosis) in mice with and without CKD. AST-120 was administered in chow before and after AV fistula creation. Administration of AST-120 significantly decreased serum indoxyl sulfate levels in CKD mice. CKD mice had a larger neointima area than non-CKD mice, and administration of AST-120 in CKD mice attenuated neointima formation. Both smooth muscle cell and fibrin components were increased in CKD mice, and AST-120 decreased both. RNA expression of MMP-2, MMP-9, TNFα, and TGFβ was increased in neointima tissue of CKD mice, and AST-120 administration neutralized the expression. Our results provided in vivo evidence to support the role of uremic toxin-binding therapy on the prevention of neointima formation. Peri-operative AST-120 administration deserves further investigation as a potential therapy to improve AV fistula patency.

## 1. Introduction

Arteriovenous (AV) fistula dysfunction, such as stenosis and thrombosis, is a major cause of morbidity and mortality among hemodialysis patients. Native AV fistulas are the best vascular access of choice owing to the low incidence of stenosis, thrombosis, or infection as compared to prosthetic grafts or central vein catheters. However, failure rate of AV fistula within two years of creation is as high as 50% [1]. In the United States, the average cost of maintaining functional vascular accesses is over 1 billion US dollars per year, and this number is on a continuous rise [2]. Nonetheless, no pharmacological agents have been proved to be effective in improving the patency of AV fistulas.

AV fistulas are usually created for end-stage renal disease (ESRD) patients for the maintenance of hemodialysis. Chronic kidney disease (CKD), especially ESRD, has been recognized as a strong risk factor for cardiovascular disease. In animal models, CKD accelerates the development of intimal hyperplasia at the venous outflow tract near the AV anastomosis [3,4]. In addition to inflammation and oxidative stress, compelling evidence suggests that the uremic milieu itself plays a critical role in the development of vascular diseases. Of note, fermentation of protein and amino acids by gut microbiota results in several toxic metabolites that are absorbed into the circulation [5]. Indoxyl sulfate (IS), p-cresyl sulfate (PCS), and indole-3 acetic acid are common uremic toxins and have been implicated in a variety of vascular dysfunction [6]. Nonetheless, the impact of these uremic toxins on AV fistula stenosis or thrombosis has not been extensively investigated.

AST-120 is an oral carbon adsorbent that can adsorb at least six negatively charged and 17 positively charged uremic toxins, including IS and PCS precursors, thereby reducing serum IS and PCS levels [7]. Previous studies showed that AST-120 could improve oxidative stress, inflammation, endothelial dysfunction, and severity of atherosclerosis [8,9,10,11]; it has been used to delay the initiation of dialysis therapy in CKD patients. Nonetheless, the effect of AST-120 on patency of AV fistula remains unknown. The study aimed to investigate the effectiveness of using AST-120 to adsorb uremic toxins to prevent neointima formation after the creation of AV fistula in CKD mice.

## 2. Results

### 2.1. CKD Mice Had Elevated Serum IS Levels Which Was Reversed Upon AST-120 Administration

The mice were divided into three groups: Non-CKD, CKD, CKD+AST-120 (Figure 1A). Serum blood urea nitrogen (BUN) levels of CKD mice were significantly elevated at 10 and 12 weeks after 5/6 nephrectomy than those of non-CKD mice (Figure 1B). Serum IS levels of CKD mice were also significantly elevated than those of the non-CKD mice at 10 weeks (32.8 ± 4.0 μg/mL vs. 6.8 ± 0.8 μg/mL, *p* = 0.0001) and 12 weeks (24.7 ± 3.6 μg/mL vs. 7.6 ± 0.8 μg/mL, *p* = 0.02) after 5/6 nephrectomy (Figure 1C). Serum IS levels of the CKD+AST-120 mice were significantly reduced compared to that of the CKD mice at either 10 weeks (19.6 ± 2.2 μg/mL vs. 32.8 ± 4.0 μg/mL, *p* = 0.02) and 12 weeks (13.8 ± 0.8 μg/mL vs. 24.7 ± 3.6 μg/mL, *p* = 0.03) (Figure 1C). There was no significant difference in body weight between the three groups of mice (Figure 1D).

### 2.2. Morphometric Analysis of the Outflow Vein Segment of AV Fistulas

The neointimal area of CKD mice was significantly increased compared to that of the non-CKD mice (281,477 ± 9850 μm^2^ vs. 209,983 ± 9885 μm^2^, average increase 35%, *p* < 0.001) at four weeks after AV fistula creation. The increase in neointima area of CKD mice was attenuated (183,900 ± 10,663 μm^2^ vs. 281,477 ± 9850 μm^2^, average decrease 35%, *p* < 0.0001) in CKD mice fed with AST-120 (Figure 2B). The IEL circumscribed area of CKD mice was significantly larger than that of the non-CKD mice (351,443 ± 14,756 μm^2^ vs. 263,769 ± 12,702 μm^2^, average increase 33%, *p* < 0.001). The increase in the IEL circumscribed area was neutralized in the CKD mice administrated with AST-120 (256,326 ± 10,079 μm^2^ vs. 351,443 ± 14,756 μm^2^, average decrease 27%, *p* < 0.001) (Figure 2C). There was no significant difference in the contribution of media thickness to whole vessel area between the non-CKD mice and the CKD mice (27.6 ± 1.8% vs. 27.1 ± 0.8%, *p* = 0.99). The patency of the AV fistula at four weeks after creation was the lowest for the CKD mice (62%) and the administration of AST-120 improved the patency rate of the CKD mice (83%), similar to that of the non-CKD mice (86%) (Figure 2D). According to the post-hoc estimation of sample size, our study was underpowered to detect patency differences because 70 mice in each group would be needed.

### 2.3. AST-120 Reduced the Smooth Muscle Cell, Collagen, and Thrombus Contents in the Neointima

Immunofluorescence statin with α-SMA, fibrinogen, and collagen antibodies was used to discriminate the change in specific compositions of neointima tissues: (Figure 3). The immunofluorescence stain demonstrated that the neointima tissues were composed of smooth muscle cells, collagen, and thrombus. Two weeks after the creation of AV fistula, the CKD mice had a larger α-SMA stained area compared to the non-CKD group (27,659 ± 1522 μm^2^ vs. 17,443 ± 1779 μm^2^, average increase 59%, *p* < 0.001). The increase in α-SMA staining area of CKD mice was reversed if the CKD mice were fed with AST-120 (18,302 ± 1208 μm^2^ vs. 27,659 ± 1522 μm^2^, average decrease 34%, *p* < 0.001).

Given that the primary thrombus was mainly composed of fibrin (nearly 60%), fibrinogen antibodies were used to evaluate the thrombus components in the neointima tissues [12]. In the venous segments near the AV anastomosis, CKD mice had a significantly larger fibrinogen-stained area as compared to the non-CKD mice (20,906 ± 1433 μm^2^ vs. 11,806 ± 1065 μm^2^, average increase 77%, *p* = 0.01). The increase in the fibrinogen-stained area of CKD mice was reversed if the CKD mice were fed with AST-120 (11,360 ± 1191 μm^2^ vs. 20,906 ± 1433 μm^2^, average increase 46%, *p* < 0.01).

The collagen content of the neointima tissue was evaluated by the Picro Sirius stain [13]. The CKD mice had an increased collagen-stained area compared to the non-CKD mice (118,282 ± 8847 μm^2^ vs. 74,175 ± 4459 μm^2^, average increase 73%, *p* = 0.01). The increase in collagen-stained area of the CKD mice was reversed if the CKD mice were fed with AST-120 (57,337 ± 10,258 μm^2^ vs. 118,282 ± 8847 μm^2^, average decrease 52%, *p* < 0.001).

### 2.4. The Effect of AST-120 on Gene Expression in AV Fistula of CKD Mice

The effect of CKD and AST-120 on gene expression in AV fistula was demonstrated on Figure 4. Two weeks after AVF creation, the gene expression of TNF-α at the juxta-anastomosis AV fistula of the CKD mice was significantly upregulated than that of the non-CKD mice (1.53 ± 0.09 vs. 1.01 ± 0.07 *p* = 0.036). The upregulation of TNF-α mRNA expression was reversed if the CKD mice were fed with AST-120 (0.68 ± 0.19 vs. 1.53 ± 0.09, *p* = 0.0005). There was no significant difference in gene expression of IL-6 and MCP-1 among the non-CKD mice, CKD mice, and CKD mice fed with AST-120.

CKD mice had an upregulated expression of MMP-9 mRNA compared to the non-CKD mice (2.789 ± 0.22 vs. 1.53 ± 0.54, *p* = 0.038), and it was attenuated by AST-120 in the CKD mice fed with AST-120 (1.11 ± 0.22 vs. 2.789 ± 0.22, *p* < 0.01). Only a trend toward increasing expression of MMP-2 in CKD mice was observed, and AST-120 also attenuated the upregulation of MMP-2 in the CKD mice.

Only a trend of increased TGF-β1 gene expression in CKD mice was observed, and AST-120 significantly surpassed the TGF-β1 gene expression in CKD mice. The expression of vascular endothelial growth factor-A (VEGF-A) in CKD mice was also surpassed if it was fed with AST-120.

There was no significant difference in the change of pro-thrombosis gene expressions, such as tissue factor and PAI-1, among the three groups of mice (data not shown).

## 3. Discussion

The present study makes a novel observation that AST-120 therapy attenuated the development of neointima after AV fistula creation. This benefit was not dependent on the change in body weight or renal function. Instead, reduction of neointima formation after AST-120 therapy was associated with a parallel decrease in plasma IS levels and mitigated pro-inflammatory cytokines and metalloproteinase in neointima tissues.

Several animal studies have shown that CKD aggravates neointimal hyperplasia in AV fistulas [3,4]. The systemic changes of CKD, such as increased oxidative stress, changes in circulating cytokines, retention of toxic metabolites, and altered calcium homeostasis may be responsible for the structural and functional changes in the maturation of AV fistulas [14]. Previous studies have shown a negative impact of oxidative stress, cytokines, and calcium metabolism on neointima formation [15,16]. Nonetheless, the role of uremic toxins on venous intimal hyperplasia was less addressed. A variety of uremic toxins play a significant role in the dysfunction of endothelial cell and vascular smooth muscle cell (VSMC). It is essential to understand the role of uremic toxins on AV fistulas because these toxins could be removed by a variety of methods. The effect of uremic toxin-binding therapy in this study supports the role of uremic toxin on neointima formation. Furthermore, our data implicated that peri-operative AST-120 might be a potential treatment to mitigate neointima formation at AV anastomosis.

Neointima formation is a complex process involving VSMC proliferation, migration, and synthesis of the extracellular matrix. There is also a close link between neointima formation and endothelium dysfunction. Although we did not explore the detailed cellular or molecular mechanisms, previous studies gave mechanical insights for the effect of AST-120 therapy. It has been demonstrated in vitro that some uremic toxins, such as IS and p-cresyl sulfate, inhibit endothelial cell proliferation and endothelial function [17,18]. Moreover, endothelial progenitor cells (EPC), which contribute to vessel repair, were decreased in CKD patients [19]. The influence of uremic toxins on EPCs was confirmed in a variety of studies [20,21]. In an ex-vivo study, uremic serum from CKD patients leads to the downregulation of contractile VSMC marker genes and increased proliferation of aortic VSMCs [22]. Normal VSMC cultured with uremic serums led to the loss of contractile phenotype markers and also an increase of extracellular matrix gene collagen 1a1, suggesting a synthetic phenotype transition [23]. Nonetheless, the effect of uremic toxins on neointima formation in vivo remained controversial [24].

Among the uremic toxins, IS the most extensively studied with regard to their negative impact on the cardiovascular systems. IS could activate the immune cells and lead to the activation of inflammatory cytokines, such as TNFα and IL-6. In animal studies, AST-120 could reduce IS-related activation of inflammatory cells and oxidative stress [10]. Uremic serum also introduces adhesion of monocytes to the endothelium, and AST-120 could mitigate the adhesion of monocytes [10]. Uremic serum could also stimulate VSMC proliferation [6]. Administration of AST-120 reduced the severity of atherosclerosis through suppression of the TGF-β and TNF-α pathway [25]. In our study, we observed an increased RNA expression of TGF-β and TNF-α in the neointima tissue. The administration of AST-120 not only lessens the neointima volume but also decreases the expression of TGF-β and TNF-α in tissues. The parallel change in the expression of proinflammatory cytokines and growth factors suggested that AST-120 might reverse the neointima formation through suppression of local inflammation.

Recently, retained uremic solutes were found to be responsible for enhanced thrombogenicity in CKD milieu in ex-vivo experiments [26,27,28]. Particularly, indolic solutes produced by tryptophan metabolism, such as IS, could increase tissue factor expression through aryl hydrocarbon receptor (AHR) signaling. The relevance of the uremic solute-AHR-thrombosis pathway on thrombosis was observed in human studies as well [29,30]. Using fibrin stain, we demonstrated that the luminal loss of AV fistula was in part secondary to thrombus formation in the vessel wall. At the venous segment near AV anastomosis, the thrombus component increased significantly in CKD mice, and it was reduced in the AST-120-treated group. Our study provided an in vivo evidence that CKD might accelerate AV fistula failure through thrombus formation at the AV anastomosis. Meanwhile, the reduction of uremic solutes using AST-120 could attenuate the formation of luminal thrombus and improve the patency of AV fistula.

In addition to inward remodeling, an unexpected finding on outward remodeling of the CKD mice was also of interest. The CKD mice had a more prominent outward remodeling than the non-CKD mice, as reflected by a larger IEL area at four weeks after creation. Administration of AST-120 inhibits the outward remodeling of CKD mice, suggesting a positive impact on outward remodeling by uremic toxins. Blood flow-induced MMP activation is a critical pathway for both the inward and outward remodeling. On the one hand, breakdown of extracellular matrix and IEL fragmentation lead to the reconstruction of the vascular scaffold, facilitating the expansion of vessels. On the other hand, breakdown of extracellular matrix also helps the migration of smooth muscle cells into the subendothelial space. The matrix fragments released by MMP-9 also activate VEGF and TGF-β production and then increase the collagen production [31,32]. In our study, upregulation of MMP-2 and MMP-9 mRNA expression was observed in the tissues of AV fistulas, in line with more outward dilatation of the venous wall, and also increased collagen deposition in the neointima. Administration of AST-20 to CKD mice downregulates MMP-9 expression in CKD mice, associated with the reversion of vessel enlargement and collagen deposition. In cultured cells, IS and PCS could increase the expression of MMP-2 and MMP-9 via the activation of the EGF receptor and downstream signaling [33]. Further studies are needed to delineate the effect that uremic serum has on outward and inward remodeling. Our results provided the net effect of inward and outward remodeling in vivo.

Several studies have investigated the effectiveness of pharmacological therapy to improve patency of AV fistula during the peri-operative period, including antiplatelets, anticoagulants, statins, fish oil, and anti-hypertensive agents. Nonetheless, systemic reviews or meta-analysis showed controversial or no beneficial effects [34,35]. Currently, no pharmacotherapy has been recommended by the guidelines to promote AV fistula maturation [36,37]. Our study demonstrated that adsorbents of uremic toxins administered in the perioperative period could attenuate neointima formation and improve patency of AV fistula. AST-120 was developed in 1982 as an oral carbonaceous adsorbent for the binding of IS and PCS precursors in the gastrointestinal tract [38]. It could effectively decrease serum IS and PCS levels in CKD patients without major side effects. Currently, AST-120 has been used for delaying the initiation of dialysis therapy in CKD patients [39,40]. Our animal study provided an evidence basis for the design of future clinical studies to investigate the effectiveness of uremic-toxin binding therapy on AV fistula maturation.

Some limitations should be addressed. Our study found that AST-120 treatment decreased circulating IS levels and lessened neointima formation. Nonetheless, the changes in IS level and neointima formation are not necessarily causally related. AST-120 is a non-specific adsorbent and a variety of uremic toxins could be removed, not only IS. Some of these toxins, such as PCS and advanced glycated end products, also have deleterious effect on vascular homeostasis. It was supported by our finding that the reduction of neointima volume and messenger RNA expression surpassed that of IS levels. Therefore, the precise mechanisms and molecular targets of AST-120 treatment still need to be clarified in future studies.

## 4. Conclusions

Our study provides in vivo evidence supporting the role of uremic toxins on stenosis and thrombosis of AV fistulas. Furthermore, the positive results in the animal study warrant further exploration of the attenuation of uremic toxin-binding therapies on neointima development after AV fistula creation.

## 5. Materials and Methods

### 5.1. Animal Experiments

All experimental procedures and protocols were pre-approved by the institutional animal care committee of Taipei Veterans General Hospital (Taipei, Taiwan), approval code: IACUC 2014-108, date of approval: 2014/08/01. Six- to 8-week-old C57BL6/J male mice were purchased from the National Laboratory Animal Center (Taipei, Taiwan). These mice were housed at 22 °C temperature, 41% relative humidity, and 12 h light-dark cycles, with access to food and water ad libitum prior to the experiments.

### 5.2. CKD Mice, Interventions, AVF Creation, and Sampling (Figure 1A)

Under intraperitoneal anesthesia with 2,2,2-tribromoethanol (Sigma-Aldrich, St Louis, MO, USA) at 200 mg/kg, both the upper and lower poles of the left kidney were removed through a lateral incision, and the cut-ends of the left kidney were cauterized to prevent bleeding. One week after the first operation, the right kidney was removed after ligation of the renal vessels and ureter under anesthesia [41]. The mice undergoing this procedure have been shown to have elevated BUN and creatinine levels at 3–4 weeks after nephrectomy. In the non-CKD group, the mice were operated on, but kidney ablation was not carried out.

The mice were divided into 3 groups: non-CKD group (shame operation), CKD group (subtotal nephrectomy), and CKD+AST-120 group (subtotal nephrectomy and 5% AST-120 treatment). One week after operation, 5% AST-120 (Kremezin, Kureha Corporation, Osaka, Japan) were mixed in pulverized chow and fed to the mice of the CKD+AST-120 group to the end of the study at 12 weeks. The mice of the non-CKD and CKD group were fed with normal pulverized chow.

Eight weeks after CKD creation, venous-end to arterial-side AV fistula was created according to the studies by Wong and Kang et al. [42,43]. The outflow vein of the AV fistulas (PV) are the dorsomedial branch of the right external jugular vein (EJV), connected to the right common carotid artery (CCA) as described. The contralateral dorsomedial branch of the left EJV of the same mouse was excised as the control vein (CV). The comparison of PV and CV at each time point would not only be due to AVF creation but also to avoid the contribution of uremia.

After AVF creation, the tissues samples were harvested on day 14 for gene expression analysis and on day 28 for histology analysis, according to the temporal change of intimal hyperplasia described by Wong et al. [43]. The AVF patency was assessed immediately after AVF creation and at the time of AVF harvesting as described previously. The tissue was processed to paraffin, and serial 5-mm sections were made perpendicular to the vein of 600 µm in length adjacent to the anastomosis.

### 5.3. Determination of Serum Blood Urea Nitrogen and IS Levels

The concentration of blood urea nitrogen and serum creatinine were determined by an autoanalyzer (SpotChem EZ, SP-4430, Arkray, Edina, USA). Supernatant in mice serum samples (50 μL) was lyophilized, and then reconstituted in 400 μL 30% acetonitrile (ACN) aqueous solution with 0.1% formic acid (FA) after protein precipitation. IS-*d_4_* (10 μg/mL) was used as an internal control and IS was detected by tandem mass spectrometry (Thermo Finnigan TSQ Quantum Ultra Mass Spectrometer, Thermo Fisher Scientific Inc., Waltham, MA, USA). The samples were sequentially injected into the UHPLC via the Acella 1250 autosampler (Thermo Fisher Scientific Inc.) and were separated using a Shiseido HPLC CAPCELL PAK C18 MGII column (150 mm × 1.5 mm, 3.0 μm, Tokyo, Japan). The mobile phases were composed of (A) 0.1% (v/v) FA in water and (B) 0.1% (v/v) FA in MeCN, with a 250 μL/ min flow rate. The MS detection mode was set with an applied voltage of 2.5 kV in the negative ion mode, and the vaporizing and capillary temperatures were set at 300 °C and 350 °C, respectively. The survey scan mode “multiple reaction monitoring (MRM)” was utilized, with MRM transitions 212 > 80 and 212 > 132 belonging to IS and 216 > 80 and 216 > 136 belonging to IS-*d_4_* for quantification. The Xcalibur software (version 2.2, Thermo-Finnigan Inc., San Jose, CA, USA) was used to acquire the MS spectra and control the mass spectrometer.

### 5.4. Quantitative Morphometric Analysis

The tissue was stained with Weigert’s elastin stain (Sigma-Aldrich, St. Louis, USA). The venous segment at 600 μm downstream to the anastomosis was sectioned at 5 μm thickness, and the most severe sections were used for the analysis [43]. After defining the internal elastic lamina (IEL) and external elastic lamina (EEL) on Weigert’s elastin stain, the area within the lumen, circumscribed by the IEL and EEL, was determined by tracing along the respective vessel regions. The media were defined as the region between the EEL and IEL, and the intima was defined as the region between the lumen ad the IEL. These measurements were performed on digitalized images using Image J software. Collagen content was determined in Sirius red-stained sections using the Image J software. This was measured in the whole vessel and expressed as the positive area (μm^2^). The AVFs were defined as patent histologically when the maximum stenosis percentage in all included venous sections was ≤95%.

### 5.5. Immunohistochemistry and Immunofluorescence Stain

Paraffin-embedded tissues were stained with H&E, and then morphometric analysis was performed. Sections were stained using primary antibodies. For the immunofluorescence stain, primary antibodies against the α-smooth muscle actin (Abcam, Cambridge, UK) and fibrinogen (Abcam, Cambridge, UK) were used to delineate the smooth muscle cell and thrombus components in the neointima area. The morphometric analysis was performed using the Image J software and expressed as positive area (μm^2^) for specific components. For immunohistochemical staining, the primary antibody against TNF-α and MMP-9 (Abcam, Cambridge, UK) was used. The sections were visualized with 3,3′-diaminobenzidine tetrahydrochloride (Dako, Glostrup, Denmark) and counterstained with hematoxylin.

### 5.6. RNA Isolation and Quantitative PCR Analysis

RNA was isolated using the NucleoZOL (Macherey-Nagel, Duren, Germany), and the quality of RNA was confirmed by the 260–280 nm ratio. The SuperScript III First-Strand Synthesis Supermix (Invitrogen, Walthan, MA, USA) was used for reverse transcription. SYBR Green Supermix (Bio-Rad Laboratories, Hercules, CA, USA) was used for real time quantitative PCR, with a 35 cycles amplification using the iQ5 Real Time PCR Detection System. Primer efficiencies were determined by the melt curve analysis. All samples were normalized by amplification of a housekeeping gene RNA. The sequences of primers used in this study are provided in the Appendix A.

### 5.7. Statistics

Statistical analysis was performed using GraphPad Prism (version 7) software. All data are expressed as means ± SEM. One-way ANOVA with post-hoc analysis was performed for morphometric, immunofluorescence, immunohistochemical, and Collagen staining in the qPCR experiment. The primary endpoint was the difference of neointima volume. According to previous studies of CKD mice, 6 mice in each group were estimated to achieve a statistical power of 80% to detect a 50% difference at an α level of 0.05 [3]. P values < 0.05 were considered significant.

## Figures and Tables

**Figure 1 toxins-12-00237-f001:**
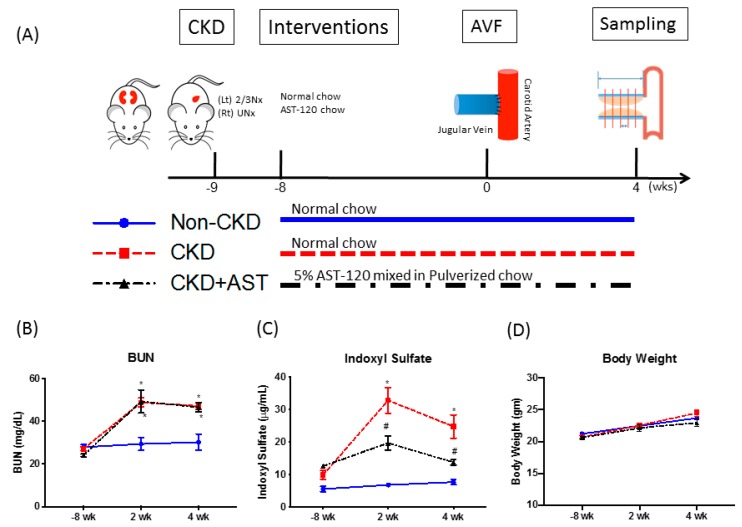
Renal function and serum indoxyl sulfate levels before and after AV fistula creation. (**A**) Mice were divided into three groups as follows: those receiving sham nephrectomy operation (non-CKD), those receiving subtotal nephrectomy with no treatment (CKD), and CKD mice treated with AST-120 (CKD+AST). (**B**) Serum BUN levels were significantly increased after subtotal nephrectomy. (**C**) Serum indoxyl sulfate level were significant increased after subtotal nephrectomy and AST-120 partially reversed the increase. (**D**) No difference in body weight was observed. **p* < 0.05 versus non-CKD group; ^#^*p* < 0.05 versus CKD group; n = 6–7 in each group. AV fistula, arteriovenous fistula; BUN, blood urea nitrogen; CKD, chronic kidney disease; 2/3 Nx, 2/3 nephrectomy; UNx, uninephrectomy.

**Figure 2 toxins-12-00237-f002:**
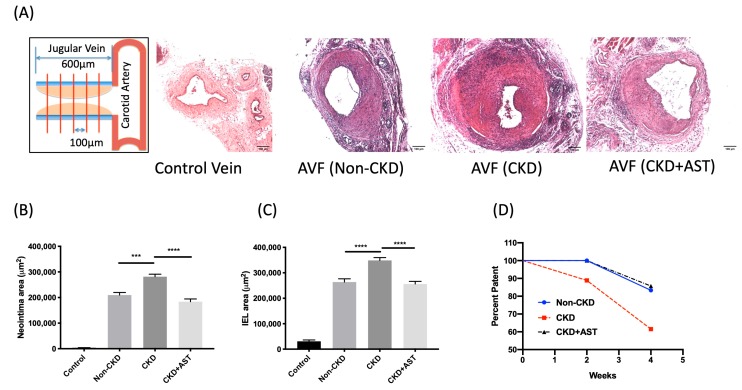
Morphometric analysis of the venous outflow track near the arteriovenous fistula (AVF) anastomosis. (**A**) The venous outflow tract at the first 600 µm distal to AV anastomosis was divided into four segments of 100-µm thickness. Tissue sections of the control vein and AVF of non-CKD mice, CKD mice, and CKD mice treated with AST-210 were stained with hematoxylin and eosin (H&E). (**B**) There were significant increases in the neointima volume of CKD mice, and the neointima volume of the AST-120-treated mice was reduced to a level similar to that of the non-CKD mice. (**C**) There was a significant increase in the internal elastic lamina (IEL) circled area (outward remodeling) in the CKD mice. The IEL area of AST-120-treated mice were reduced to a level similar to that of the non-CKD mice. (**D**) Kaplan Meier estimates demonstrate that CKD mice had lower AVF patency than the non-CKD mice at 4 weeks after creation. CKD mice had a lower patency rate than the non-CKD mice. The patency of CKD mice treated with AST-120 at 4 weeks after creation was improved (AST-120 vs. CKD, 83% vs 62%, *p* = 0.15, n > 6 at each time point). ****p* < 0.001; n = 6–7 in each group.

**Figure 3 toxins-12-00237-f003:**
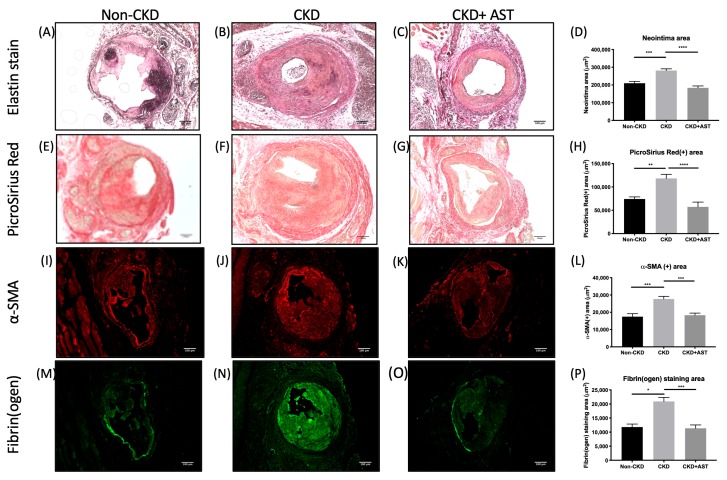
AST-120 decreased both smooth muscle cell, collagen, and thrombus contents in the neointima of AV fistulas. Both elastin stain (**A**–**D**) and PicroSirius Red stain (**E**–**H**) showed an increased AVF neointima area in CKD mice, and the increase was counteracted by AST-120 at 4 weeks after AVF creation. Immunofluorescence staining of α-SMA (**I**–**L**) and fibrin(ogen) (**M**–**P**) revealed a larger positively stained area in the AVF neointima of CKD mice than in the non-CKD and AST-120-treated CKD mice at 2 weeks after AVF creation. **p* < 0.05, ***p* < 0.01, ****p* < 0.001, **** *p* < 0.0001. Data presented as mean ± SEM. Data analyzed by one-way ANOVA followed by Bonferroni post hoc analysis, and n = 6–7 in each group. *Scale bar*: 100 μm.

**Figure 4 toxins-12-00237-f004:**
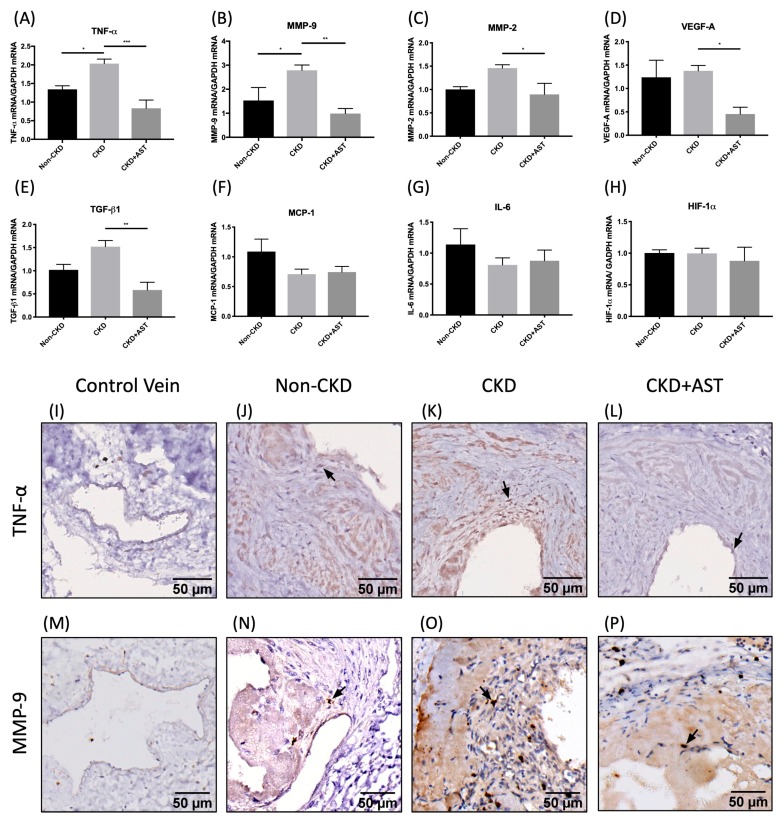
AST-120 neutralized the upregulation of TNFα and MMP-9 in CKD mice. The expression of mRNA at the venous segment of AV fistula showed upregulation of TNF-α mRNA (**A**) and MMP-9 mRNA (**B**) in the neointima of CKD mice, which was reversed when AST-120 was administrated to the CKD mice. The mRNA expression of MMP-2 (**C**), VEGF-A (**D)**, and TGF-β1 (**E**) in the AVF venous segment was suppressed by AST-120 treatment. The mRNA expression of MCP-1 (**F**), IL-6 (**G**), and HIF-1α (**H**) was not significantly different among the non-CKD mice, CKD mice, and CKD mice treated with AST-120. Immunohistochemical staining of TNF-α (**I**–**L**) showed a larger intensely stained area in the neointima of CKD mice comparing to Non-CKD and CKD+AST mice and control vein. The immunohistochemical staining of MMP-9 (**M**–**P**) showed a similar pattern as that of TNF-α. * *p* < 0.05, ** *p* < 0.01, ****p* < 0.001, **** *p* < 0.0001. Data presented as mean ± SEM. Data analyzed by one-way ANOVA followed by Bonferroni post hoc analysis, and n = 6–7 in each group. ANOVA, analysis of variance; AVF, arteriovenous fistula; CKD, chronic kidney disease; HIF-1α, hypoxia-inducible factor α; IL-6, interleukin-6; MCP-1, monocyte chemoattractant protein-1; MMP-2, matrix metalloproteinase-2; MMP-9, matrix metalloproteinase-9; PAI-1, plasminogen activator inhibitor-1; TGF-β1, transforming growth factor-β1; TNF- α, tumor necrosis factor-α; VEGF-A, vascular endothelial growth factor-A.

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
