# Peer review of "Oral Charcoal Adsorbents Attenuate Neointima Formation of Arteriovenous Fistulas"

_toxins, 2020, doi:10.3390/toxins12040237_

Round 1

Reviewer 1 Report

The present study examines a clinically highly relevant topic, i.e. AV fistula stenosis in patients with chronic kidney disease, and interesting first data are presented on the potential effects of a relatively simple intervention to prevent this complication.

Overall, the manuscript is well written (see minor comments below) and the data are presented in a concise manner.

I have the following criticism which may help to improve the study.

Main comments:

  • Why were mice analyzed at 10 and 12 weeks, i.e. with only a 2 week difference?
  • Representative images and results shown in Figure 2A and 2B are the same as the ones shown in Figure 3A and 3D. That is not possible and one of both must be removed. A different stain could be used to analyze elastic fibers in Figure 3A and 3B or to determine neointima area in Figure 2A and 2B.
  • The anti-SMA stain in Figure 3I to 3K does not look convincing, one would expect more SMCs within the neointima. Was a positive control used during immunostaining?
  • Upregulation of MMP9 would be expected to result in increased matrix degradation and reduced fibrosis. How do these data match with the observed increased collagen in Figure 3I to 3L? Please discuss.
  • The discussion starts with the statement that effects of AST-120 were independent of changes in body weight or kidney function. Where are these data shown?
  • The number of mice examined per group is quite low (n=6-7). Was the number of mice per group predefined based on expected end points? Please comment and discuss.
  • The main „problem“ of the study is the absence of experiments demonstrating causility, such as analysis of smooth muscle cell proliferation and migration in response to IS with and without AST-120. As of now, it could be that two or more independent changes are observed that are not causally linked. Ideally, the authors could provide such experiments. Otherwise, this limitation must be clearly mentioned and alternative possibilities discussed.
  • Methods: the time point and duration of AST-100 treatment and mode of application is not clearly described in the Methods. How long before the AV fistula surgery was it given or afterwards? How and how often was it applied?

Minor comments:

  • Numbers in brackets should be given in the same order as the groups are described in the text (e.g first the results of the CKD and then the non-CKD mice).
  • Please indicate the number of mice used for analyses shown in Figure 4.
  • Please check again for typographical errors (e.g omission of space before unit in lines 56 and 59).
  • Labeling of mice in Figure 1A does not match with the labeling of the group in the text and also is not explained or not consistent with the explanation.

Reviewer 2 Report

This study examines AV fistula remodeling in a mouse model of chronic kidney disease (CKD) following renal mass reduction to determine the effectiveness of an oral charcoal absorbent (AST-120) in decreasing the uremic toxin indoxyl sulfate (IS) levels and preventing fistula neointima formation. Results suggest that treatment with AST-120 prior to creation of an AV fistula could ultimately improve patency.

It is not clear why morphological measurements were quantified 4 weeks following AV fistula creation while the expression levels of pro-remodeling protein and mRNA were determined 2 weeks following surgery. Also, at two week it is not clear if IS is significantly elevated in the CKD+AST mice (text and figure are inconsistent).

If uremic toxins are a primary contributor AV fistula remodeling, why does AST-120 lead to only a partial reduction in IS levels (CKD-AST compared to non-CKD mice) but a complete attenuation of neointima formation and changes in protein/mRNA levels?

Did fistula medial area change in CKD could changes in medial area be a contributing factor to the outward remodeling in CKD?

Round 2

Reviewer 1 Report

The authors have responded to all of my comments and I have no further suggestions.

Reviewer 2 Report

No major comments at this time.

One typo noted in the revised text on Page 2, line 58 "were significantly reduced than that of the CKD mice"